# Universal, school-based interventions to promote mental and emotional well-being: what is being done in the UK and does it work? A systematic review

Karen Mackenzie,[1] Christopher Williams[2]

[1]Psychological Services NHS Ayrshire and Arran, Ayrshire Central Hospital, Irvine, UK
[2]Mental Health and Wellbeing, University of Glasgow, Glasgow, UK

**Correspondence to**
Dr Karen Mackenzie;
karen.mackenzie@aapct.scot.nhs.uk

## ABSTRACT

**Objectives** The present review aimed to assess the quality, content and evidence of efficacy of universally delivered (to all pupils aged 5–16 years), school-based, mental health interventions designed to promote mental health/well-being and resilience, using a validated outcome measure and provided within the UK in order to inform UK schools-based well-being implementation.

**Design** A systematic review of published literature set within UK mainstream school settings.

**Data sources** Embase, CINAHL, MEDLINE, PsycINFO, PsychArticles, ASSIA and Psychological and Behavioural Sciences published between 2000 and April 2016.

**Eligibility criteria** Published in English; universal interventions that aimed to improve mental health/emotional well-being in a mainstream school environment; school pupils were the direct recipients of the intervention; pre-post design utilised allowing comparison using a validated outcome measure.

**Data extraction and synthesis** 12 studies were identified including RCTs and non-controlled pre-post designs (5 primary school based, 7 secondary school based). A narrative synthesis was applied with study quality check.[1]

**Results** Effectiveness of school-based universal interventions was found to be neutral or small with more positive effects found for poorer quality studies and those based in primary schools (pupils aged 9–12 years). Studies varied widely in their use of measures and study design. Only four studies were rated 'excellent' quality. Methodological issues such as small sample size, varying course fidelity and lack of randomisation reduced overall study quality. Where there were several positive outcomes, effect sizes were small, and methodological issues rendered many results to be interpreted with caution. Overall, results suggested a trend whereby higher quality studies reported less positive effects. The only study that conducted a health economic analysis suggested the intervention was not cost-effective.

**Conclusions** The current evidence suggests there are neutral to small effects of universal, school-based interventions in the UK that aim to promote emotional or mental well-being or the prevention of mental health difficulties. Robust, long-term methodologies need to be pursued ensuring adequate recording of fidelity, the use of validated measures sensitive to mechanisms of change, reporting of those lost to follow-up and any adverse

## Strengths and limitations of this study

► Addressed a gap in the literature.
► Used a robust methodology to review the literature in this area.
► Conclusions will help inform UK policy and practice as this topic continues to be debated in current health, education and political spheres.
► Included papers largely based in England so unlikely to be representative of the cultural diversity within UK schools.
► Date limit excluded papers published prior to 2000 and after April 2016. There were insufficient resources to update the literature search beyond this timepoint prior to publication.

effects. Further high-quality and large-scale research is required across the UK in order to robustly test any long-term benefits for pupils or on the wider educational or health system.

## INTRODUCTION

The mental and emotional well-being of children and young people has received increasing attention worldwide. It has been reported that the prevalence of mental health problems ranges from 10% to 20%[2] and that by the age of 18 years up to 20% of young people will have experienced an emotional disorder.[3] Mental health conditions such as anxiety and depression often persist into adulthood[4] and have been associated with a range of negative outcomes including lower academic achievement, higher likelihood of health risk behaviours, self-harm and suicide.[5 6] However, provision of services for those in need can be as low as 20%.[7] Such access issues to specialist services like Child and Adolescent Mental Health Services (CAMHS) has meant that school-based interventions have been increasingly explored due to their far reach[8] and existing infrastructure to support

child development[9] while noting that schools need support to use the evidence base when applying such interventions.[10]

Numerous systematic reviews and meta-analyses have been conducted to review the effectiveness of school-based mental health interventions at both the universal (delivered to all pupils irrespective of perceived need) and targeted (delivered to vulnerable or 'high risk' individuals only) levels. Overall, this literature has indicated mixed results regarding efficacy of school-based interventions.

Findings have suggested positive effects on social emotional skills, self-concept, positive social behaviours, conduct problems, emotional distress and problem solving when reviewing school-based universal programmes aiming to enhance social and emotional skills.[11 12] Further reviews found cognitive behavioural therapy (CBT) formed the basis of the majority of anxiety prevention programmes (78%) and over 75% of trials reported a significant reduction in anxiety.[13] CBT-based interventions were also tentatively endorsed as mildly effective in reducing depression (Effect size [E.S.].=0.29) and moderately effective (E.S.=0.50) in reducing anxiety symptoms.[14]

With regards to optimal implementation, it has been noted that more positive outcomes were obtained for programmes adopting a 'whole-school' approach that lasted more than 1 year and aimed to promote mental health rather than prevent mental illness.[12] A balance of both universal and targeted approaches has been recommended, along with accurate implementation of interventions.[15]

However, the long-term impact and target audience of such initiatives has been questioned. A meta-analysis reviewing prevention of depression programmes found that while there was evidence of immediate postintervention effects, these did not sustain over time (24–36 months).[16] Moreover, a review evaluating both anxiety and depression programmes found that while the majority were effective for depression (65%) and anxiety (73%), the effect sizes were small (0.12–0.29).[17]

It has also been argued that universal prevention interventions are, overall, not efficacious,[18 19] with targeted programmes being most effective (E.S.=0.21 to 1.40). Likewise, that while school-based CBT programmes have been demonstrated to lead to a short-term reduction in depression symptoms, interventions are most effective for those in the clinical range.[20]

The literature has, therefore, conveyed conflicting results regarding the efficacy of universal school-based interventions while consistently highlighting methodological issues within the existing research base. Common issues include a lack of active intervention controls;[21] studies' operationalisation and measurements of 'resilience' lacking homogeneity[22]; that weak programme fidelity and treatment dosage impacts outcomes[11]; and that there is insufficient use of validated, standardised measures and long-term follow-up.[23]

It is also noteworthy that the majority of reviews have focused worldwide, with most reviewed interventions based in Australia, the USA or Canada. No reviews to date have focused solely on studies in schools in the UK. This trend was also referenced in a National Institute for Health and Care Excellence (NICE)-funded review[24] of targeted and universal school-based interventions who noted that though findings from international based research are helpful, the generalisability to the UK educational system is questionable. Education system differences between countries and continents such as funding, political drivers, curriculum pressures and workforce planning issues give rise to a need for reviews specifically within the UK context, especially while local funders and UK commissioners face calls to address rising mental health problems in schools. Therefore, it is particularly timely to have access to the most relevant information drawn from the current literature as it pertains to the UK educational system specifically.

One systematic review of targeted school-based interventions within the UK research has been conducted.[25] This found that nurture groups demonstrate an immediate positive impact on the social and emotional well-being on vulnerable young people; however, results from longer term follow-up studies are less clear.

The need to carry out a review of universal school-based interventions specifically within the UK context therefore remains. This is especially pertinent in light of the increasing emphasis from national government on developing CAMHS services within the UK, and the impetus on health and education services to work together in order to improve well-being outcomes for children and young people.[26–28]

## Review aims
The present review aims to fill this gap in the literature by focusing on universally delivered, school-based mental health interventions provided within the UK only. The following questions will be explored:
1. How effective are universal school-based interventions in the UK that promote mental health, emotional well-being or psychological resilience and what tools are being used to measure effectiveness?
2. What methodologies are being applied in UK schools when trialling interventions and what is the quality of these studies?
3. What are the intervention characteristics, for example, delivery, content and target audience?
4. What are the identified barriers in delivering and evaluating universal school-based interventions?

## Search strategy
Electronic databases were searched for relevant published research on 14 April 2016: Embase, CINAHL, MEDLINE, PsycINFO, PsycArticles, ASSIA and Psychological and Behavioural Sciences. Selected journals relevant to the area were hand-searched (*British Journal of Educational Psychology* and *British Journal of School*

*Nursing*). Previous reviews and relevant papers were reviewed, and following consultation with university librarians, keyword search terms were identified and linked with the Boolean operators 'AND' and 'OR' (see online supplementary file for search strategy examples).

Study design criteria were wide to allow for the diverse range of methodologies used to overcome challenges in school-based research. Search terms were, therefore, chosen primarily to promote sensitivity to the subject area. A limit date was set from 2000 to April 2016. The early date limit was selected as this area has been promoted by UK governmental policy largely within the last decade. Furthermore, detailed appraisal of the previous systematic reviews in this area found few, if any, discovered studies prior to this date.

### Study selection
The inclusion criteria were as follows:
► The intervention was based in a mainstream school environment.
► The intervention was universal in its application (ie, to all pupils irrespective of need).
► Pupils were the direct recipients of the interventions.
► The study adopted a pre-post design.
► The intervention aimed to target mental health and/ or emotional well-being.
► The study used a validated measure to quantitatively evaluate emotional or mental well-being outcomes and reported those outcomes.
► The study was published in English between 2000 and April 2016 in a peer-reviewed journal.

Exclusion criteria included
► The study aims or methodology did not fit the inclusion criteria.
► Any studies using a non-validated outcome measure as their primary outcome, for example, Likert scales that were unvalidated.
► Any studies using a purely qualitative methodology.

### Details of included and excluded studies
Duplicate papers were excluded. Titles were screened to identify only those that clearly met inclusion criteria. Abstracts were assessed independently by the authors. Raters met to compare included papers. Where eligibility was unclear based on the abstract, full articles were retrieved and assessed jointly by raters. Reference lists of included papers were searched as well as previous reviews on related topics. Articles citing included articles were also reviewed, and one paper was sourced via this method. Authors of protocol papers were contacted leading to an additional paper being sourced. Experts in the field in Scotland, England, Northern Ireland and Wales were contacted regarding any other studies. However, none were eligible for inclusion. Twelve papers were included in the final review (see figure 1).

### Quality rating of studies
The Downs and Black[1] checklist was used to assess quality. This checklist assesses internal and external validity, selection bias and study power over 27 items. This checklist was used due to its utility in assessing studies relating to public health and its applicability to assess quality in both randomised and non-randomised studies. Reliability and validity assessment has found the quality index to have high internal consistency, good test–retest ($r=0.88$) and inter-rater ($r=0.75$) reliability and good face and criterion validity (0.90).[1]

A sample of papers were assessed by an independent researcher (CA). Any rating discrepancies were discussed and a shared decision reached. A decision was taken not to exclude any studies found to be of poor quality as the aim of this current review was to critique universal school-based interventions while acknowledging that the real-world implementation of such evaluations can be challenging and, as a result, may reasonably impact study quality.

### Data extraction
Due to the heterogeneity of the studies, meta-analysis was not appropriate. A narrative synthesis was applied to explain the findings of this review in line with current guidance.[29] Information gathered from the studies included: study aim, intervention (model, duration and delivery), sample characteristics, study procedures, outcomes and measures, and results. Issues relating to the implementation, as well as effectiveness, of interventions were also noted from those studies commenting on such barriers.

### Patient and public involvement
No patients or members of the public were directly involved in this piece of research.

## RESULTS
### Overview of interventions
Of the 12 studies sourced, five took place in primary schools[30–34] and seven took place in secondary schools.[35–41] An overview of study interventions based in primary and secondary schools can be found in table 1.

### Primary school studies
The five studies within primary school settings evaluated interventions based on computerised CBT[30]; a teacher-led intervention embedded within the curriculum (eg, 'Promoting Alternative Thinking Strategies' (PATHS)[31]); manualised anxiety interventions (eg, a locally developed anxiety intervention or the Australian developed 'FRIENDS' programme) delivered by both school staff (teachers and nurses) and external health staff (eg, psychologists).[32–34]

### Secondary school studies
Three of the secondary school-based studies trialled interventions based on CBT principles (eg, UK Resilience Programme (UKRP) and Resourceful Adolescent

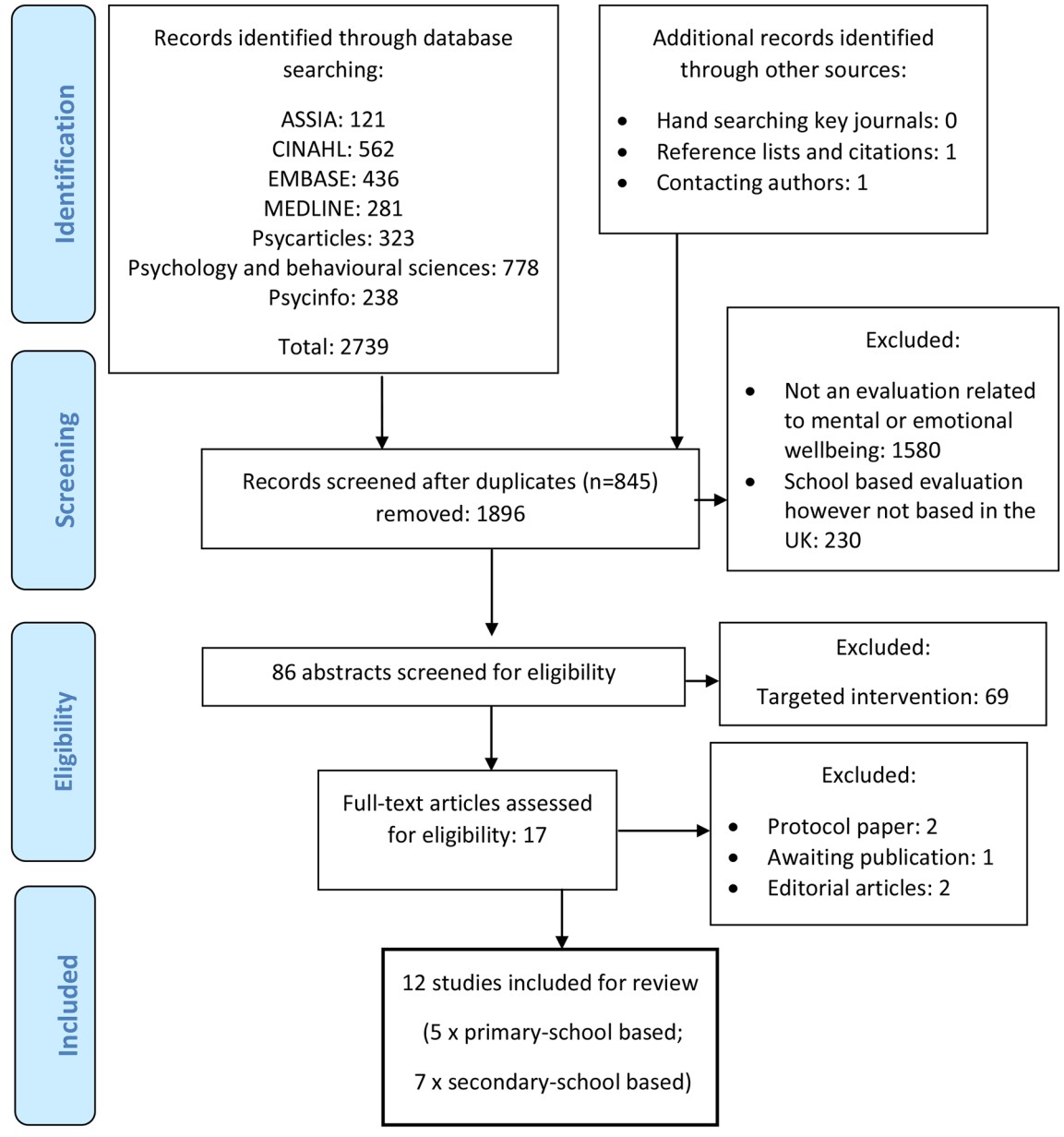

**Figure 1** PRISMA flow diagram. PRISMA, Preferred Reporting Items for Systematic Reviews and Meta-Analyses.

Programme (RAP-UK)[36 39 41]) delivered by school staff,[36] educational psychologists[39] and external facilitators.[41] Interventions were also said to include principles of interpersonal therapy (RAP-UK[41]) and behavioural approaches (Thinking about Reward in Young People ('TRY'[39])).

One study trialled an intervention based on positive psychology,[35] two studies trialled a mindfulness-based intervention[38 39] and two trialled locally developed mental health education sessions delivered to all pupils.[37 40] These interventions were led by trained school teachers[35 38 40] and trained volunteers.[37] All delivered the intervention during Personal Health and Social Education (PHSE) classes.

### Methodological quality
The quality of studies ranged from 'poor' (34%[30]; 37.5%[35]) to 'excellent' (75%[34 37]; 78.1%[36]; 81.3%[41]).

Six studies used a randomised controlled pre-post design.[30–32 34 37 41] The remaining were non-randomised pre-post designs, and only one did not have a control group.[33] Some studies were particularly weak on their description of sample characteristics and representation of the population,[30 35] reporting of those lost to follow-up and accounting for those in the analysis,[32 35] and the exploring of adverse events, of which only one study provided information.[41] Only six studies provided a power calculation,[31 34 36 37 40 41] most of which had samples sufficiently powered to determine an effect (except ref [37]). The remaining studies did not provide such information.

Of the 11 studies employing controls, six used controls from the same school in which the intervention was taking place.[32 34 36 37 39] All other studies recruited controls from different schools.

**Table 1** Overview of interventions based in primary and secondary schools

| Study (location) | Sample | Study aim/hypothesis | Intervention – theoretical model and content | Intervention – setting, structure and delivery |
|---|---|---|---|---|
| **Primary schools** | | | | |
| Attwood et al[30] (Bristol, England) | Boys aged 10–12 years from two coeducational schools (n=13). | A proof of concept study to explore the viability and possible benefits of a computerised CBT (cCBT) programme. | 'Think, Feel, Do' – based on CBT principles with a psychoeducation component. Cartoon characters guide users through various activities including: emotional recognition; linking thoughts, feelings and behaviours; identifying and challenging negative thoughts; and problem solving. Involves quizzes, practical exercises, videos, music and animation. | Six x 45 min sessions delivered via an interactive multimedia CD-ROM. Took place within the school and facilitated by the researcher. |
| Berry et al[31] (Birmingham, England) | Pupils aged 4–6 years (n=5075; 56 x schools). | Test the effectiveness and cost-effectiveness of the intervention to reduce children's level of behavioural and emotional difficulty. | 'Promoting Alternative Thinking Strategies' aims to improve skills in five areas: self-awareness, managing feelings, motivation, empathy and social skills. Lessons are developmentally sequenced and focus on techniques for self-control; emotional and interpersonal understanding; steps for solving interpersonal problems; positive self-esteem; and improved peer relationships. | 44 lessons in year 1; 47 lessons in year 2. Delivered by trained teachers within classroom. Manual provides teacher scripts, pictures, activity sheets, photos, posters and home activities. |
| Collins et al[32] (South Lanarkshire, Scotland) | Pupils aged 9–10 years (n=317; 9 schools; 18 classes). | To explore if anxiety and coping showed improvement postintervention, and test effects of delivery. | 'Lessons for living: Think Well, Do Well'. CBT-based intervention to develop coping skills. A series of skills practice using interactive teaching methods. Children are guided to recognise emotional symptoms, reduce avoidant coping strategies and focus on proactive problems solving and support-seeking. | Ten lessons delivered by a psychologist (n=103) and teacher (n=79) during PHSE. Teachers provided with intervention manual following 1 day training. |
| Stallard et al[33] (Bath and North East Somerset, England) | Pupils aged 9–10 years (n=106; three schools; four classes). | To evaluate an Australian-originated intervention in the UK; test delivery by school nurses. | 'Feelings, Relax, I can do it, Explore solutions, Now reward, Don't forget practice, Smile' (FRIENDS). Based on CBT principles, it teaches children practice skills to: identify their anxious feelings and learn to relax; identify unhelpful thoughts and replace them with helpful thoughts; face and overcome problems and challenges. | Ten sessions delivered by school nurses who attended 2-day training. Lessons comprise group work, workbooks, role play and games. Parents invited to preintervention session. |
| Stallard et al[34] (Bath, North East Somerset, Swindon, Wiltshire, England) | Pupils aged 9–10 years (n=1448; 45 × schools). | To assess the effectiveness of FRIENDS delivered by both health and school professionals on anxiety prevention. | As above.[33] | Nine × 60 min lessons delivered to whole classes. Health-led group: two trained facilitators. Teacher-led group: led by class teacher. All attended 2-day training. |
| **Secondary schools** | | | | |
| Boniwell et al[35] (South East London, England) | Pupils aged 11–12 years (n=296; 2 x Haber-dashers' Aske's Fed. of Schools) | To test the efficacy of a new school programme for the promotion of happiness and well-being skills. | 'Personal Wellbeing Lesson Curriculum'. Covers the 'scientific basis of happiness' focusing specifically on two core aspects: positive emotions/experiences and positive relationships. Based on theoretical constructs from well-being research and positive psychology for example, 'three good things', forgiveness letter and gratitude visit. | Eighteen biweekly 50 min scripted lessons delivered to eight classes by four teachers who attended a 5- day training. Provided with lesson plans, PowerPoints and handouts. |
| Challen et al[36] (Greater London, North West England and North East England) | Pupils aged 11–12 years (n=2844; 16 × schools) | To evaluate a UK version of Penn Resiliency Program. Hypothesised high completion rates and reduction of depression symptoms. | 'UK Resiliency Program'. Aims to build resilience and promote realistic thinking and adaptive coping based on Ellis's 'Activating event-belief consequences model'. Teaches cognitive behavioural and social problem-solving skills; encourages accurate appraisal of situations; and assertiveness, negotiation and relaxation skills. | An 18-hour programme delivered within the timetable at the teacher's discretion. Delivered by school staff who attended a 10-day training in the USA. |

Continued

**Table 1** Continued

| Study (location) | Sample | Study aim/hypothesis | Intervention – theoretical model and content | Intervention – setting, structure and delivery |
|---|---|---|---|---|
| Chisholm et al[37] (Birmingham, England) | Pupils aged 12–13years (n=769; 6 × schools). | To test whether contact with an individual with Mental Health (MH) diagnosis plus education is more effective in reducing stigma, improving MH literacy and promoting well-being than education alone. | 'Schoolspace'. A 10-module MH intervention designed by study researchers covering topics such as stress, depression, psychosis, different ways of thinking and a drama workshop. The 'contact' group had an individual facilitating who was an MH service user and had a diagnosis (eg, psychosis and Bipolar Disorder); this was revealed halfway through the day. | A 1-day intervention within the school led by National Health Service staff, trained volunteers and MH service users. |
| Kuyken et al[38] (England) | Pupils aged 12–16years (n=522; 12 × schools). | To investigate the acceptability of a mindfulness programme for teachers and students; test efficacy of programme on MH and well-being. | 'Mindfulness in Schools Program' (MiSP). Involved learning to direct attention to immediate experience with open-minded curiosity and acceptance. Skills were learnt through practice sessions and everyday application. Mindfulness practice used to work with mental states and everyday stressors to cultivate well-being and promote mental health. | Nine weekly scripted lessons delivered as part of the curriculum, or at lunchtime by seven teachers trained and approved to deliver the MiSP curriculum. |
| Rice et al[39] (South East England) | Pupils aged 13–14years (n=256; 3 × schools). | To compare three types of intervention that may prevent adolescent depression and explore cognitive mechanisms involved with each. | 'Thinking about Reward in Young People' (TRY) aimed to enhance reward processing through actively selecting activities to lift mood. CBT aimed to change negative thinking patterns by encouraging evaluation of thoughts. Mindfulness Based Cognitive Therapy aimed to promote awareness and acceptance of thoughts and to develop regulation of attention through guided meditation. Psychoeducation regarding depression was provided to all groups. | Eight weekly manualised sessions of each intervention delivered within 50 min PHSE lessons by educational psychologists who attended regular supervision. |
| Naylor et al[40] (Greater London, England) | Pupils aged 14–15years (n=416; 2 × schools). | To explore whether teaching adolescents about mental health would result in gains in knowledge and empathy. | Mental health lessons. Topics included: stress, learning disability, depression, suicide/self harm, eating disorders and bullying using methods such as discussion, role playing and internet searching. | Six x 50 min weekly lessons delivered by seven group tutors from pastoral care who attended a 1- day training from researchers. |
| Stallard et al[41] (Bath, North East Somerset, Bristol, Wiltshire, Nottinghamshire, England) | Pupils aged 12–16years (n=5030; 8 × schools, 28 × year groups). | To assess effects of classroom-based CBT on symptoms of depression and in relation to other aspects of psychological well-being and specific demographic subgroups. | 'RAP-UK: Resourceful Adolescent Programme'. A depression prevention programme based on CBT and interpersonal therapy principles adapted to fit the UK curriculum. Key elements include: personal strengths, helpful thinking, keeping calm, problem solving, support networks and keeping the peace. Students complete workbooks as they progress. | Nine x 50–60 min manualised lessons delivered within the PHSE curriculum by two trained facilitators external to the school. Two booster sessions offered to schools at 6-month follow-up. |

CBT, cognitive behavioural therapy; PHSE, Personal Health and Social Education.

Sample sizes ranged from 13[30] to 5075.[31] The age of participants ranged from 4[31] to 16 years old[38 41] with the majority of studies targeting the early adolescent age range (9–12 years old) at the end of primary school or at the beginning of junior/secondary school.[30 32 34–37]

## EFFECTIVENESS OF INTERVENTIONS

An overview of study characteristics and outcomes can be found in tables 2 and 3.

### Data collection and measurement

Studies varied widely in their use of measures. Measures used to rate depressive symptoms included the Children's Depression Inventory (CDI),[36] the Short Mood and Feelings Questionnaire (SMFQ)[39 41] and the Centre for Epidemiological Studies – Depression Scale (CES-D).[38] Measures used to rate anxiety included the Revised Children's Anxiety and Depression Scale,[34 41] Revised Children's Manifest Anxiety Scale,[36] Penn State Worry questionnaire[41] and the Spence anxiety scale.[30 32 33] Measures used to capture different methods of coping related to symptoms of anxiety or depression included: Children's Automatic Thoughts Scale,[41] Coping Strategy Indicator,[32] Sentence Completion for Events in the Past Test[39] and Perceived Stress Scale.[38] Two studies used measures related specifically to well-being or resilience: Warwick-Edinburgh Mental Well-being Scale (WEMWBS)[38] and the Resilience Scale,[37] and others used measures related to self-esteem[33 34 41] and life satisfaction.[35] The Strength and Difficulties Questionnaire (SDQ) was the most commonly used measure said to rate behavioural, emotional difficulties and overall functioning, and either the child, parent or teacher version was used in 6 of the 12 studies.[30 31 33 36 37 40] Studies varied according to the length of follow-up ranging from 4 weeks[37] to 2 years.[41] Four of the 12 studies sought to obtain qualitative, as well as quantitative data.[30 35 37 41] However, it was beyond the scope of this paper to comment on qualitative findings.

Due to the heterogeneity of studies, the effectiveness of each intervention approach will be discussed in turn. Overall, results suggested a trend whereby higher quality studies reported less positive effects.

### Studies trialling bespoke mental health education programmes (n=3[35 37 40]; – all in secondary schools).

Two studies found small (d=0.11–0.22) but significant improvements in total and subscale SDQ scores for those that received mental health education. However, of those, it is noteworthy that Chisholm et al[37] did not employ a non-intervention condition. Boniwell et al[35] trialled a bespoke intervention based on positive psychology principles and found a decrease in outcomes of life satisfaction and an increase in negative affect for both groups. However, this was less so for the intervention group (d=−0.24 compared with d=−0.79), which was interpreted as the intervention having a 'buffering effect' at a time of stress for the pupils.

### Studies trialling CBT-based interventions (n=8; 30–34,36,39,41). These are described by setting (primary and then secondary).

#### Primary schools

All primary-school based studies trialled interventions pertaining to altering thinking styles based on CBT principles. Four studies, three of which employed a control arm, reported statistically positive outcomes on anxiety-related measures following interventions including FRIENDS,[33 34] 'Think Feel Do'[30] and locally developed CBT programmes[32] with larger effects for those in 'high risk' groups (d=−1.26[33]; no control arm). Methodological issues such as a small sample size and significant group differences at baseline (n=13[30]), failure to include those lost to follow-up in analysis,[32] lack of controls[33] and small effect sizes for universal samples (d=0.01–0.2)[34] should be noted when taking inference from those results. Mixed results were found in relation to delivery, with stronger effects found in interventions led by health professionals (d=0.2) versus school staff (d=0.02),[34] or no difference between psychologist or teacher-led interventions.[32] A sufficiently powered, good quality study evaluating the use of PATHS within the curriculum found few, small significant results (d=0.06–0.14; teacher-rated intervention measure) at 12-month follow-up and no effects on any measure at 24-month follow-up.[31]

#### Secondary schools

Fewer significant outcomes were found in trials based within secondary school populations. Small (d=0.093) but short-lived positive outcomes were found on the CDI for those in the UKRP intervention.[36] Mixed results were found for those in the RAP-UK intervention, with results indicating some beneficial and also potentially negative outcomes[41] although all with small effect sizes. Both were high-quality, longitudinal, well-powered studies employing robust methodologies. Furthermore, no effects were found in the CBT group when compared with as-usual controls or other treatments in a smaller study looking at mechanisms of change.[39] In the same study, a behavioural intervention (TRY) was found to have positive effects on reward-seeking behaviour and SMFQ measure (d=−0.8) when compared with other treatments; however, this finding was not confirmed when compared with PHSE-as-usual controls.

### Studies using mindfulness-based interventions (n=2[38 39]; - both in secondary schools).

Positive outcomes were found in a feasibility study evaluating a mindfulness-based intervention,[38] which yielded statistically significant, modest effects on both depression (CES-D: d=−0.24) and well-being (WEMWBS: d=0.15) measures. Due to small sample sizes, this study was likely to be underpowered; however, outcomes were sustained at 3-month follow-up and were associated with greater mindfulness practice. No significant outcomes were found in a smaller study trialling MBCT on measures of mood (SMFQ) or reward-seeking.[39]

**Table 2** Design and outcome characteristics of primary-school based studies

| Study (% quality rating) | Study design | Measures | Follow-up | Effects/outcomes |
|---|---|---|---|---|
| Attwood et al[30] (34%) | Randomised pre-post intervention evaluation using opportunistic sample. No blinding or randomisation procedure reported. 'cCBT' (n=6) × control group (n=7). | ▶ SCAS – Parent & Child version. ▶ SDQ – parent version. ▶ Focus groups (n=8). | Baseline; 6 weeks postintervention. | Significant reduction in SCAS-C 'social' (d=0.49*) and 'general anxiety' (d=0.48*) subscales (note: intervention group significantly higher on SCAS at baseline). No effects on parent rated measures. |
| Berry et al[31] (68.8%) | Randomised controlled trial; web randomisation system. 29 schools 'PATHS' intervention × 27 schools WL control.† | ▶ SDQ – teacher version. ▶ PATHS teacher rating scale (PTRS). ▶ T-POT. | Baseline; 12-month postintervention; 24-month postintervention. | No differences on SDQ at 12-month follow-uup. Some significant results on subscales of PTRS at 12-month follow-up (social competence: d=0.09*; aggression: d=0.14*; inattention: d=−0.06*; peer relations: –0.10*). Not maintained at 24-month follow-up. |
| Collins et al[32] (46.9%) | Randomised 3×3 mixed design. No randomisation procedure reported. Psychologist-led anxiety intervention (n=103) × teacher-led anxiety intervention (n=79) × controls (n=135). | ▶ CSI ▶ SCAS – Child version administered by teachers. | Baseline; postintervention; (within 3 weeks of end); 6-month follow-up. | Improvement in psychologist-led and teacher-led groups on SCAS-C (d=0.41*; d=0.31*) and CSI 'Avoidance' (d=0.31*; d=0.31*) and 'problem solving' (d=−0.66*; d=0.52*) subscales. No difference between psychologist or teacher-led groups. SCAS-C outcomes maintained at 6-month follow-up (d=0.39*; d=0.39*). Noted: those lost to follow-up (n=155) were not included in analysis. |
| Stallard et al[33] (43.4%) | Pre-post evaluation of pupils (n=106) from three schools taking part in the FRIENDS intervention. No controls employed. | ▶ SCAS-Child version. ▶ CFSEQ. | 'T1': 6 months prior; 'T2': prior to intervention; 'T3': 3-month follow-up. | Improvements in SCAS (d=−0.50*) and CFSEQ (d=0.58*) from T1 to T3 for whole sample; not between T2 and T3 (across intervention). Improvements on both measures (d=−1.26*; d=−1.27*) for 'high risk' group between T2 and T3. |
| Stallard et al[34] (75%) | Cluster randomised controlled trial randomised through computer tool. Health-led FRIENDS (n=489) × school-led FRIENDS (n=472) × controls (n=401).† | ▶ RCADS 30 – child & parent. ▶ Penn State Worry Questionnaire. ▶ RSES. ▶ Bully/victim questionnaire. ▶ Subjective well-being assessment. ▶ SDQ – Parent version; teachers completed 'Impact scale'. | Baseline; 6-month follow-up; 12-month follow-up. | Improvement on total RCADS (d=0.20*) and social (d=−0.09*) and general anxiety subscales (d=−0.20*) – not depression. Smaller effect sizes in school-led group (d=0.02*; d=0.11*; d=0.01*). No statistical improvements on secondary outcome measures or teacher/parent rating scales. |

*Significant at p<0.5 level.
†Study sufficiently powered to detect change.
CFSEQ, Culture-Free Self-esteem Questionnaire; CSI, Coping Strategy Indicator; PATHS, Promoting Alternative Thinking Strategies; PTRS, PATHS teacher rating scale; RCADS, Revised Child Anxiety and Depression Scale; RSES, Rosenberg Self-Esteem Scale; SCAS, Spence Children's Anxiety Scale; SDQ, Strength and Difficulties Questionnaire; T-POT, Teacher Pupil Observation Tool.

**Table 3** Design and outcome characteristics of secondary school-based studies

| Study (% quality rating) | Study design | Measures | Follow-up | Effects/outcomes |
|---|---|---|---|---|
| Boniwell *et al*[35] (37.5%) | Non-randomised control group pre-post design. 'Personal Wellbeing' intervention group (n=211) × control group (n=85). | ▶ SLSS. ▶ MSLSS. ▶ PNASC. ▶ Qualitative interviews. | Baseline; postintervention (10-month follow-up). | No significant improvement on SLSS or MSLSS. Decrease in 'satisfaction with school' (d=0.4*) and 'friends' (d=−0.17) scores for whole sample. Decrease in positive affect for both intervention and control groups (d=−0.24*; −0.79*); increase in negative affect (d=0.54*) for control group. Noted: those lost to follow-up (n=103) not accounted for in analysis. |
| Challen *et al*.[36] *(78.1%)* | Non-randomised pragmatic controlled trial. UKRP intervention (n=1016) group × control (n=1894) group.† | ▶ CDI. ▶ RCMAS. ▶ SDQ. | Baseline; postintervention (4–9 months); 1-year follow-up; 2-year follow-up. | Small significant impact on CDI postintervention (d=0.093*); not maintained at 1-year or 2-year follow-up. No significant effects on RCMAS or SDQ scores. |
| Chisholm *et al*[37] (75%) | Pragmatic cluster randomised controlled trail, randomised by independent researcher. 'Contact and MH Education' (n=354) group × MH education (n=303) group.‡ No 'as usual' controls. | ▶ RIBS (not validated for adolescents). ▶ MAKS (not validated for adolescents). ▶ SDQ. ▶ Resilience scale. ▶ Helpseeking Q. ▶ Focus groups. | Baseline – 2 weeks prior to intervention day; 2-week postintervention day. | Statistical sig. improvements on several scales postintervention day for both groups – 'contact and education' and 'education only': attitudinal-based stigma (d=0.23*; d=0.25*), knowledge based stigma (d=0.54*; d=0.59*), mental health literacy (d=0.05; d=0.13*) emotional well-being (d=0.16*; d=0.14*) and resilience (d=0.07; d=0.22*). No change in 'helpseeking'. |
| Kuyken *et al*.[38] *(59%)* | Non-randomised controlled feasibility study. MiSP intervention group (n=256) × control (n=266). | ▶ WEMWBS. ▶ PSS ▶ CES-D. ▶ Mindfulness practice. | Baseline; postintervention (9 weeks); 3-month follow-up. | Lower depression scores postintervention (d=−0.29*). Improvement on all measures at 3-month follow-up (WEMWBS: d=0.15*; PSS: d=−0.09*; CES-D: d=−0.24*). Mindfulness practice significantly associated with greater gains across all measures (unable to calculate E.S.). |
| Rice *et al*[39] (50%) | Non-randomised longitudinal design with three intervention conditions. TRY intervention group (n=50) × CBT group (n=53) × MBCT group (n=54) × PHSE controls (n=99). | ▶ SMFQ. ▶ CGT to measure reward seeking. ▶ DASC and corresponding response time. ▶ SCEPT to measure overgeneral memory. | Baseline; 9-week follow-up. | Statistical sig. changes in reward seeking in TRY group (d=0.12*); no change after CBT or MBCT. No statistically significant decrease in SMFQ across groups compared with PHSE controls. When comparing treatment groups only, TRY showed statistical reduction in SMFQ when compared with MBCT and CBT (d=−0.8*); reward-seeking moderated reductions in SMFQ scores (d=1.62*). |
| Naylor *et al*[40] (56.3%) | Non-randomised pre-post control group study. MH intervention group (n=175) × control group (n=242).† | ▶ Mental Health Questionnaire (unvalidated). ▶ SDQ. | Baseline (1 week before intervention); 6 months postintervention. | Improvement in MHQ with regards to awareness of depression causes (d=0.21*) and bullying (d=0.31*). Changes in specific SDQ subscales: 'conduct' (d=0.22*) and 'prosocial' (d=0.11*) but not on total difficulties. |

Continued

**Table 3**   Continued

| Study (% quality rating) | Study design | Measures | Follow-up | Effects/outcomes |
|---|---|---|---|---|
| Stallard *et al* (2013) (81.3%) | Cluster randomised controlled trial, randomised by computer. RAP-UK intervention group (n=1753) × attention controls (n=1673) × PHSE controls (n=1604).† | ▶ SMFQ.<br>▶ CATS.<br>▶ RSES.<br>▶ RCADS.<br>▶ School connectedness.<br>▶ Attachment questionnaire.<br>▶ European Quality of Life-5 dimensions. | Screening – SMFQ only; baseline; 6-month follow-up; 12-month follow-up. | No significant effect on SMFQ at 12-months follow-up. Some effect of intervention on bullying status at 12 months, and cannabis use at 6-month and 12-month follow-up. Intervention less useful than usual PHSE or attention controls for panic; less useful than usual PHSE on CATS 'personal failure' and general anxiety. Signs of benefits and harm of intervention found, all reported to be small effect sizes (data unavailable to calculate effect size). |

*Significant at p<0.5 level.
†Study sufficiently powered to detect change.
‡Power calculation provided but proportion lost to follow-up (>15%) reduced sample required for adequate power.
CATS, Children's Automatic Thoughts Scale; CBT, cognitive behavioural therapy; CDI, Children's Depression Inventory; CES-D, Centre for Epidemiologic Studies Depression Scale; CGT, Cambridge Gambling Task; DASC, Dysfunctional Attitudes Scale for Children; E.S., effect size; MAKS, Mental Health Knowledge Schedule; MBCT, Mindfulness-based Cognitive Therapy; MSLSS, Multidimensional Students Life Satisfactions Scale; PHSE, Personal Health and Social Education; PNASC, Positive and Negative Affect Schedule for Children; PSS, Perceived Stress Scale; RCADS, Revised Children's Anxiety and Depression Scale; RCMAS, Revised Children's Manifest Anxiety Scale; RIBS, Reported and Intended Behaviour Scale; RSES, Rosenberg Self-Esteem Scale; SCEPT, Sentence Completion for Events in the Past Test; SDQ, Strength and Difficulties Questionnaires; SLSS, Student's Life Satisfaction Scale; SMFQ, Short Mood and Feelings Questionnaire; UKRP, UK Resilience Programme; WEMWBS, Warwick-Edinburgh Mental Well-being Scale.

### Implementation issues

Common issues relating to implementation were found across all studies.

### Fidelity

Fidelity to intervention delivery was highlighted as an issue in terms of both measurement and outcome. Studies used self-rated fidelity methods,[32] external fidelity ratings on a proportion of sessions[31 34 36 37 41] or no fidelity rating methods reported at all. Studies commented variably on the possible effect of fidelity and 'treatment dosage' on outcomes. In Stallard *et al*'s[37] study, the health-led condition with 100% fidelity (ie, administered all pieces of homework and activity tasks), was associated with significantly better outcomes than the school-led group who achieved 60%–80% fidelity. 'High quality' workshops were also found to be related to greater declines in CDI measures.[36] Conversely, Berry *et al*[31] found that fidelity (when applying an arbitrary '80%' rate of 'high' fidelity) was not found to be related to outcome.

### Attrition

Investment from schools was raised as an issue as demonstrated by school participation and attrition[31 41] and failure to administer follow-up measures as per study procedures.[32 35] All studies, with the exception of Stallard *et al*,[41] provided little information about school or participant characteristics of those who dropped out. This confounding factor may have positively biased results. For instance, in Kuyken *et al*'s[38] study, teachers who delivered the mindfulness intervention had been invested in the intervention for approximately 2 years before the beginning of the study and attended regular supervision, demonstrating good motivation throughout the study that found positive outcomes.

### Costs

Two studies actively explored health economic costs involved.[31 41] Cost-effectiveness was not calculated by Berry *et al*[31] due to lack of impact, and Stallard *et al*[41] concluded that the intervention was not cost-effective. Of note, both studies may have sustained high costs due to employing external facilitators to lead the intervention rather than teachers[41] and hiring 'coach consultants' to monitor delivery.[31]

### DISCUSSION

This review aimed to explore the effectiveness and study quality of universally delivered school-based interventions within the UK that aim to promote mental health and emotional well-being. Several clear conclusions can be drawn from this review, while other issues require further clarity from future research.

### How effective are universal school-based interventions in the UK that promote mental health, emotional well-being or psychological resilience?

Based on the studies included in this review, the effectiveness of universal school-based interventions remains mixed and, at best, modest. Where there were several positive outcomes, effect sizes were small and methodological issues rendered many results to be interpreted with caution. This prudent finding echoes the somewhat mixed results from worldwide reviews,[11–24] where while several positive evaluations exist, this finding is not

consistent when applied across diverse settings and populations, which calls into question the overall generalisability of school-based interventions in the literature to real-world environments.

Notwithstanding, this current review focusing solely on UK schools found that studies based in primary schools seemed to find more encouraging results from CBT-based interventions on measures of anxiety, although most studies had methodological limitations relating to use of appropriate controls and failure to include those lost to follow-up in analysis. Positive results tended to fall in the older age range of primary school pupils (9–12 years old).

Within the secondary school population, the most positive results were obtained when delivering mental health education sessions, behavioural or mindfulness interventions. Two high powered, good quality studies evaluating CBT-based interventions within secondary populations found few significant results, and one study indicated possible detrimental impacts of the intervention compared with controls, although any effect sizes related to these findings were small.

It is curious that studies fail to detect promising effects in the older, secondary school, population. It could be argued that a 2-year follow-up is not sufficient to truly detect change or prevention during the developmentally sensitive time that is adolescence. Arguably, the demands placed on adolescents merely change in nature rather than impact over time. Adolescent psychosocial development[42] is particularly vulnerable as individuals are required to manage academic demands as they progress through their school career, navigate friendships, seek to develop self-identities and deal with the physiological changes that occur as they transition through puberty. It could be that the existence of such pervasive and fluctuating stressors juxtaposed with measurement issues, discussed below, contribute to the failure to detect significant results in secondary school populations. Or, that such interventions simply have less impact for this population.

### What methodologies are being applied in UK schools when trialling interventions and what is the quality of these studies?

Methodological issues were predominant in this review. Only four of the studies were of 'excellent' quality, and findings indicated a trend towards higher quality papers finding fewer positive results. Studies were weakened largely due to their lack of randomisation and blinding of researchers, and small sample sizes that likely rendered them underpowered to detect true effects.

While it was encouraging that initial consenting rates were high and remained reasonable throughout, study quality would benefit from better reporting of those lost to follow-up who, possibly, could be a population of particular interest when considering the objective of promoting mental and emotional well-being for all within the school setting. Furthermore, statistical methods used to account for such missing data require careful consideration to ensure that more stringent and conservative methods — for example, intent-to-treat analyses — are applied in school-based research. Otherwise, studies that instead apply a 'defined completers' or 'completers' analysis expose themselves to the risk of yielding false positives.

Another issue was the use of controls. Few studies explicitly provided details of the content controls groups received. Some indicated that controls may have already received materials available in the school around social and emotional well-being, which could reasonably have confounded results. Additionally, considering the demographic data provided, it is unlikely that the included studies accurately represent the cultural diversity of schools across the UK; therefore, caution should be taken when considering the generalisability of results.

The last prominent issue highlighted in this study was the diverse use of measures and length of follow-up across studies, making it difficult to ascertain a coherent picture of measurement and effects in the current research base.

As commented in one study[36] and further afield,[22] measurement issues within universal populations are particularly problematic due to common floor effects, particularly when using measures pertaining to the existence of mental health conditions. As has been well documented, demonstrating improvement in 'high risk' groups is somewhat easier as baseline scores are often elevated providing scope for reduction.[41] Demonstrating change within a universal population is therefore inherently more difficult and requires careful thought when moving forward. Is it sufficient that the absence of a mental health condition equates to greater well-being or resilience as suggested by Boniwell et al,[35] or should researchers direct attention to explicitly measuring well-being and resilience and mechanisms of change within such constructs in order to truly operationalise factors relating to the prevention of mental health difficulties?

Few studies in this review used well-being or resilience measures. However, those that did[37 38] found positive effects. While any meaning of these results must be taken with caution due to methodological issues, this nevertheless suggests that such measures are at least able to detect change within a universal population.

Only one study explored mechanisms of change[39] by using cognitive reasoning tests when comparing several interventions and found that a behavioural intervention led to more reward seeking and a reduction in mood symptoms. It would be of value to explore this further given the neurodevelopmental stage of early adolescence when frontal lobes are still maturing and neuronal connections continue to grow.[43] Consequently, the adolescent's ability to plan, problem solve and manipulate abstract information, as is arguably necessary in cognitive-based interventions, may be over-ridden by more disinhibited, emotionally driven impulses and the seeking of concrete rewards, as may be seen in earlier adolescence and would potentially explain increased receptiveness to a behavioural rather than cognitive intervention.

It could also be of value that future studies take a more holistic perspective of general well-being during evaluation of universal populations. Such indicators may include school attendance, exam completion, referrals rates to local CAMHS, academic outcomes, long-term mental and physical health outcomes, occupational or further education uptake, as well as important qualitative components.

### What are the identified barriers in delivering and evaluating universal school-based interventions?

Implementation barriers relating to fidelity to intervention delivery and costs were also raised within this review. Variance in fidelity measurement to confirm reliable manualised delivery was a recurring issue, which is of particular salience when delivery has been consistently argued to be related to outcome.[11 13] Intervention delivery itself varied between studies where school staff or external researchers delivered the courses. While results were mixed when comparing the effectiveness of teacher-led versus externally led interventions, overall within this review, the results were neutral suggesting, at best, that there is no negative impact of teacher delivery. While issues relating to treatment fidelity may be more prominent with teacher delivery, considering sustainability, it could be argued that this would be the optimal approach in school settings, especially considering the financial costs involved in employing external facilitators as demonstrated by two studies in this review.[31 41] Furthermore, research has indicated that pupils prefer both that mental health education be delivered by someone with a thorough knowledge of the subject and for it to be delivered by someone they know, for example, a teacher.[44]

No study in this review explored the impact on any allied services such as CAMHS. For instance, it may be useful to audit local CAMHS referral rates while reviewing the effectiveness of school-based interventions, and whether an increase or decrease in referrals would be observed. Considering the absence of reliable positive outcomes at the individual level at this point, a systemic perspective could be of value when considering any cost benefits to the wider health and social care services.

Furthermore, it was unclear from the review what local or national political or strategic drivers instigated each study, and indeed, the extent to which children and young people were consulted in the process, design and delivery of the interventions. It was outside the scope of this review to explore the qualitative findings from the few studies that employed focus groups. Therefore, it is recommended that future qualitative reviews of school-based research are conducted in order to ensure that children's and young people's views as stakeholders in this work are sufficiently represented.

### Limitations

This study was limited in its ability to source evaluations representative of the entire UK as the majority of studies were based in England. While efforts were made to source evaluations from elsewhere in the UK, the lack of validated measures or application of pre-post methodology meant that such evaluations from the 'grey literature' could not be included in this review. It should therefore be noted that there is much relevant work being conducted in schools across the UK. However, schools and local authorities should be urged to reliably evaluate their valuable efforts and contribute to the published literature, thereby demonstrating the important work being driven by teachers and policymakers nationwide.

This study was also limited in its date source in that only studies from the year 2000 were included in this review. While results from other systematic reviews suggested that little relevant research was done in the UK before this time, it could still be that some studies were missed due to this limit.

### Implications

This review highlighted the need to employ robust methodological designs within school-based research in order for any effects to be interpreted meaningfully. Measurement issues exist where they do not adequately detect change in universal populations, and there is a wide variety of measures used ranging from 'clinical' to well-being measures. This review concludes that school-based researchers across the UK should attempt to come together to discuss ways to address this issue and improve coherence in the literature.

An additional, imperative implication from this review is the proactive inclusion and involvement of teachers in this work. As has been commented elsewhere[45] without the 'buy-in' from teachers, any school-based intervention is less likely to sustain or achieve positive outcomes. In a time of additional pressures on teachers, the need to feel in control of initiatives is key. Of note, two of the studies in this review included adult-focused exercises for the teachers themselves as an adjunct to the intervention training. This approach may go further to assist teachers' stress management and understanding of mental health while attending to the needs of their pupils.

### CONCLUSIONS

The current evidence suggests there are neutral to small effects of universal, school-based interventions in the UK that aim to promote emotional or mental well-being or prevention of mental health difficulties. While the real-world limitations of conducting research in schools exists, robust, long-term methodologies need to be attempted when conducting research in this area in order to explore the longitudinal impact of school-based interventions on well-being. Academic attainment, school attendance and rates of high-risk presentations also need to be further explored. This requires adequate recording of fidelity, the use of validated measures sensitive to mechanisms of change, reporting of those lost to follow-up and any adverse effects and the use of qualitative data to supplement quantitative outcomes. Interventions in the existing UK-based literature include educational, behavioural, cognitive and mindfulness components,

each demonstrating variable results. Nevertheless, national and local policy[26–28 46] indicates that there remains an appetite to develop work in this area in order to promote well-being outcomes for children and young people. In this case, further research collaborations are required across the UK to robustly demonstrate any benefits for pupils or on the wider system.

**Acknowledgements** Many thanks to Dr Claire Adey (CA) who assisted in the quality rating process.

**Contributors** This piece of research was submitted in part fulfilment for a doctorate degree in clinical psychology with the University of Glasgow. KM was the main researcher and responsible for developing the research questions, conducting the search strategy and analysing results. CW supervised this research and acted as co-rater during the search process.

**Funding** The authors have not declared a specific grant for this research from any funding agency in the public, commercial or not-for-profit sectors.

**Competing interests** CW is immediate past president of the BABCP, the lead body for CBT in the UK, and a CBT researcher and trainer. He is also the author of a range of CBT-based resources including some aimed at primary and secondary school populations. These are available commercially as books, online courses and classes. He receives royalty and is shareholder and director of a company that commercialises these resources.

**Patient consent** Not required.

**Provenance and peer review** Not commissioned; externally peer reviewed.

**Data sharing statement** Supplementary data available on request to the author.

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
