## [Reviewer comments · BMJ Open]

ARTICLE DETAILS

TITLE (PROVISIONAL)	Universal, school-based interventions to promote mental and emotional wellbeing. What is being done in the UK and does it work? A systematic review
AUTHORS	Mackenzie, Karen; Williams, Christopher

VERSION 1 – REVIEW

REVIEWER	Franco Veltro Department of Mental Health, ASReM
REVIEW RETURNED	12-Apr-2018

GENERAL COMMENTS	The systematic review has been conducted without bias. However we suggest to more emphasize works and researchs in this field with particular attention to Emotional Intelligence.
--

REVIEWER	Sarah Atkinson Durham University, UK
REVIEW RETURNED	25-Apr-2018

GENERAL COMMENTS	The paper offers a systematic review of studies examining the benefits of universal (ie not targeted) interventions in schools in the UK (primary and secondary up to 16years of age) that aim to improve mental health, as assessed through diverse instruments. The strategy for searching for studies is clearly described, the criteria for inclusion in the review clear and rigorous, and the description of the final selection of 12 papers sufficiently detailed and clear. The authors take a narrative synthesis to presenting the results given the wide range of intervention form and evaluation tools. The findings are very clearly presented, including summaries in a table format. In addition to reviewing the results of the interventions, the authors also summarise related aspects of adherence to the intervention programme, drop-out and characteristics of the drop-outs (very underreported in most studies), and aspects of the costs of the interventions (again only discussed in a few studies). The review returns to the study questions set out at the beginning of the paper and directly addresses each in turn. Limitations are discussed and some preliminary implications drawn. The review is of interest, particularly given the evidence of very limited benefits gained from such interventions and, moreover, that the better the quality of the evaluation, the less significant such benefits are. The lack of consistency in measures used for evaluation in this field is highlighted. The paper is well presented and well expressed throughout with clear tables and figures. The paper is a pleasure to read and draws out some interesting points. I have a few relatively minor queries and suggested amendments or
---

	additions:  1. The paper states early on that 'Numerous systematic reviews and meta-analyses have been conducted' on this (p3). However, only one has been undertaken in the UK. The rationale, therefore, for this review is implicitly that it is important to have knowledge that is county, setting or education system specific. The argument for why this is important, however, is never explicitly developed. 2. As a follow-up to the rationale for the study, we might have expected some return to considering how the findings in the UK relate to other findings elsewhere, which appeared to have been rather more positive, albeit mixed. And what does this tell us about potential generalisability of the interventions, given the use of what is, after all, to a large extent an intentionally 'decontextualised' method (through case-control, pre-post, evaluations ?) 3. Previous findings had suggested that impact might be greater amongst those with mental health challenges in the clinical range. How can the benefits of such interventions on preventing deterioration, that is sustaining mental health levels (resilience?) rather than actively showing an improvement, ever be assessed in this way? There are a couple of grammatical errors: p7. 188 Criteria were (criteria =plural) p8 221, none was (none=singular) p8. 248 No patients or members of the public was (No = singular)
--	---

REVIEWER	Judi Kidger University of Bristol, UK
REVIEW RETURNED	30-Apr-2018

GENERAL COMMENTS	This paper presents a clear account of recent UK school-based interventions to address mental health in children and young people. It also includes a reasonable critique of the quality of the studies. However I have two major concerns with the paper that make it unsuitable for publication in its current form: the first is that it does not really add anything to current knowledge. There are a number of similar systematic reviews in this area already, and the better quality studies in this review have been included in other studies. The only reason some of the studies in this review have not been included in previous reviews is because of their poorer quality. Therefore by limiting this review to studies based in the UK, all the authors have done in effect is repeat previous reviews, but included a poorer selection of studies. If the review had contained more of a focus on what is feasible in the UK context then maybe it would have been clearer why this review was needed, but there isn't any discussion about the UK context in particular. Therefore the conclusions - that we can't really conclude anything about the effectiveness of interventions in the UK due to poor study quality - is not really a surprise to anyone familiar with the field. My second major concern is that it is very unclear what the aims and the inclusion/exclusion criteria actually were. At some points in the paper it appears the review is of studies that focus on wellbeing and resilience, so for example under the aims the authors state that they are including interventions that "promote mental health, emotional wellbeing, or psychological resilience". However in other places and looking at the search strategy and papers actually included, it is clear that the review wasn't just about studies that promoted wellbeing, but it was also about studies that prevented depression or anxiety. For example in the discussion page 23 line 414 the authors say they were looking at interventions "which aim to promote mental and
--

	emotional wellbeing, or prevent mental ill health". The concern here is that if the inclusion/exclusion criteria weren't clearly defined a priori, then it casts doubt on the systematic nature of the review. Relatedly, it is not clear from the inclusion/exclusion criteria whether the authors were only including studies that involved classroom based teaching, as opposed to interventions that changed the school environment. All the included studies are psychoeducational interventions, but it is not clear if this was because they were the only studies that were found, or because they were the only studies that were included. A final point on the exclusion/inclusion criteria is that the authors should state here that they only included universal interventions - this is stated elsewhere so surely was an inclusion criterion? One more, less major point, is that the authors include a section about the barriers that exist to delivering the interventions, but the evidence they produce doesn't really address this question. The authors state that any qualitative data from the studies was beyond the scope of the review, but in fact the qualitative data are what would have answered this question the best. In fact if the review had included qualitative findings that examined things like barriers and feasibility in the UK context, then I believe it would have made more of a contribution to current knowledge.
--	---

VERSION 1 – AUTHOR RESPONSE

Reviewer 1:

We welcome the comments from reviewer 1 that the study has been conducted well and without bias.

There is a suggestion that we introduce links to the wider literature on emotional intelligence as a predictor of subjective wellbeing. Although of wider interest, we feel this dilutes the specific focus on the current paper which did not evaluate this.

Reviewer 2:

We were pleased that the focus of the review is praised as being clear and relevant. The study processes are described as rigorous and clear, with a very clear presentation of results. We welcome the acknowledgement that our work has contributed by identifying drop-out and characteristics of the drop-outs post-intervention- something currently under-reported in the literature. The lack of consistency in measures, and identification that higher quality studies lead to poorer outcomes are highlighted as key points. We were delighted the reviewer found the paper a delight to read. The following changes have been made:

- 1). The argument that UK-specific systematic review has been expanded on page:5 line 147.
- 2). Comments have been added to link our findings in the UK, with the wider more impactful results identified in key world-wide reviews page 22, line 44.
- 3) We agree with the reviewer's comments regarding the challenge in evaluating the effectiveness of interventions for those in the non-clinical range. This has been addressed further on Page 25, line 131.
- 4). The grammatical errors identified have been corrected where agreed. However the suggested grammatical changes around using the words "none" or "no" are contested, as their use in both contexts is in plural form which negates the need to change the associated verbs.

Reviewer 3:

Reviewer 3 was far more negative about the paper. Specific issues are:

1). The paper does not add anything to current knowledge. The reviewer points out there are existing international systematic reviews in this area, and the papers identified in our study are included. There is a concern the focus on UK studies has reduced the quality of the study.

We have focused on the UK-based research in order to help inform local policy makers, funders and schools as to the current state of the literature as it relates to the UK system only. This is to appeal to local readership and ensure the cultural relevance and direct impact of the paper to real-life environments, which is of utmost importance to the author. We have addressed this in greater detail as per our response to reviewer 2 above. We feel this significantly enhances the rationale and need for the current paper.

2). Clarity of the inclusion and exclusion: we have clarified information already previously in the paper that the inclusion criteria addressed both mental illness ('anxiety' and 'depression') as well as wellbeing/resilience. This was decided, a priori, in detailed consultation with the University librarians following review of the wider literature which used such terms. This somewhat wide criteria with regards to those terms was decided upon to ensure we would capture sufficient works in our search, whilst maintaining a narrow, strict, scope with regards to methodology (e.g. pre-post design,; validated measures). This is now clearly stated in the inclusion/exclusion criteria (Page 7; line 205), and we have also added a supplementary file with the detailed search strategies used. Finally, we have also completely re-written the abstract in line with the Prisma abstract checklist to add clarity to the process. To clarify, the studies were all chosen to be classroom based delivery.

3). We confirm the review focused on Universal interventions. We have been clear that the criteria included only universal classroom based studies.(Page 7, line 201)

4). The reviewer makes the point that study should have also reviewed qualitative findings. Unfortunately this was a student-based study with no funding and which had to be done to a high quality in a fixed period of time. The review fulfils its aims well, but we feel a qualitative review was beyond the scope planned.

VERSION 2 – REVIEW

REVIEWER	Sarah Atkinson Durham University
REVIEW RETURNED	05-Jul-2018
GENERAL COMMENTS	My few concerns with the first version of the paper were largely points of clarity. These have all been addressed well and I would be happy for the paper to be published in its current form.